# Utilization of Modified Red Mud Waste from the Bayer Process as Subgrade and Its Performance Assessment in a Large-Sale Application

Shijie Ma, Zhaoyun Sun *, Jincheng Wei, Xiaomeng Zhang 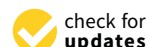 and Lei Zhang

Shandong Transportation Research Institute, Ji'nan 250102, China; mashijie@sdjtky.cn (S.M.);
weijincheng@sdjtky.cn (J.W.); zhangxiaomeng@sdjtky.cn (X.Z.); realchungsdu@163.com (L.Z.)
* Correspondence: andersontwo@163.com

**Abstract:** The utilization of red mud waste discharged from the Bayer production process used for extracting alumina from bauxite presents a pressing demand in the aluminum industry. This study aims to adopt a chemical modifier to solidify the Bayer red mud for its application in highway subgrade. The mechanism and properties of the modified red mud using a modifier composed of cement, phosphogypsum and organic polymer, were analyzed and investigated. It was found that the optimal modifier dosage of the solidified modifier was 8%. The three-day unconfined compressive strength of the modified Bayer red mud could reach up to 3 MPa and its strength loss when immersed in water at 7 days and 28 days measured less than 20%. For its real application as subgrade, its road performance could be achieved with good bearing capacity, including a resilient modulus value greater than 90 MPa, a dynamic deformation modulus reaching up to 140 MPa and the Falling Weight Deflectometer (FWD) value measuring less than 100 (0.01 mm). Compared with traditional lime or cement stabilized soil, using locally modified Bayer red mud for subgrade filling can reduce the project cost, minimize the consumption of non-renewable resources and reduce the emission of environmental hazards, thus providing an engineering reference for large-scale and resource-based road applications.

**Keywords:** red mud waste; subgrade; modification; road performance



## 1. Introduction

Red mud is an industrial solid waste discharged from the production process of extracting alumina from bauxite, with the Bayer process representing a widely used alkaline alumina production process [1,2]. Approximately 1.0 to 1.8 tons of waste residue is discharged per ton of alumina produced as a byproduct of the Bayer process, with the waste referred to as Bayer red mud. With the rapid development of the aluminum industry in China, the amount of discharged Bayer red mud continues to increase. At present, the alumina production industry generally stores Bayer red mud through a dry-heaping embankment technology [3]. However, its open-air storage occupies a large amount of land, which continuously consumes management and maintenance costs, and more importantly, brings risks and hidden dangers to the surrounding ecological environment [4,5].

Many studies have attempted to implement the comprehensive utilization of Bayer red mud and offer technical solutions for recycling of the production of industrial building materials [6–8], extraction and recovery of valuable metals [9–11], production of new materials [12,13], and environmental restoration [14–16]. However, Bayer red mud still cannot be used in large-scale operations thus far, due to mutual constraints of various factors such as technical and economic feasibility, and secondary pollution. Therefore, the cumulative stock of red mud in China has exceeded 350 million tons, but its overall comprehensive utilization rate remains less than 4% [17].



In practice, it is acknowledged that the construction of roads, railways and other infrastructure projects require large amounts of soil and rock for subgrade filling, which not only wastes a large amount of non-renewable land resources, but also causes increasingly significant damage to the environment and ecology. Using Bayer red mud as a roadbed filling material for engineering construction through systematic research can alleviate the shortage of soil sources during the construction process and realize the large-scale consumption and utilization of bulk solid waste in the aluminum smelting industry, which has important practical significance.

Bayer red mud is alkaline, contains trace amounts of chromium, fluoride and other harmful components, and has characteristics of high natural moisture content, fine particles, and poor water stability. Therefore, it cannot be directly used as subgrade filling material from the perspective of environmental protection and engineering applications. Over the years, extensive work has been conducted by researchers worldwide to develop various utilization methods of red mud in road engineering. E Mukiza et al. prepared an eco-friendly road base material using mainly red mud and characterized its mechanical properties, hydration and leaching characteristics [18]. CVH Rao carried out unconfined compressive strength, splitting tensile strength and California bearing ratio tests on different amounts of GGBS stabilized red mud, and concluded that red mud stabilized with GGBS can be used as subbase, base and subgrade [19]. V Mymrin et al. used red mud, clay slate mining wastes and polishing sludge to prepare composite materials, resulting in the improvement of all its mechanical properties, and which could be used in road foundations, airport runways and building foundations [20]. J Qi et al. carried out the preparation and curing test of red mud road base material based on the analysis of the physical and chemical properties of red mud. Subsequent test results showed that the red mud base exhibited better compressive strength and resilience modulus than the traditional road base material [21]. Previous laboratory studies indicated that red mud was feasible for use as a raw material in road base or subgrade by adopting advanced technology/treatment methods. These studies mainly focused on the selection of red mud curing materials, mix design and improvement of mechanical properties, but research on improving the utilization rate of red mud, the control of leaching pollutants and real road performance remain inconclusive.

In this study, based on the analysis of the production process, the chemical composition and engineering and environmental properties of original Bayer red mud, its pollution source, environmental risks and road characteristics were first assessed. Furthermore, a chemical modifier was adopted to solidify the Bayer red mud, attain effective inhibition of the leaching of harmful components in red mud and to achieve the utilization rate of red mud above 90%. In a real highway project, a large-scale engineering application and road performance of the modified Bayer red mud as subgrade filling were evaluated. In this paper, the key technologies of road performance improvement and environmental risk control were comprehensively considered for the modification treatment of red mud, and the modified material with red mud as the main body was used for expressway subgrade filling for the first time. The road and environmental performance were monitored and evaluated by the physical project, providing a valuable reference for the large-scale application of red mud in road engineering.

## 2. Materials and Methodology

### 2.1. Original Red Mud Waste from the Bayer Process

2.1.1. Chemical Composition of Bayer Red Mud Waste

The chemical composition of Bayer red mud waste, collected from a local aluminum industry, is shown in Table 1. The Bayer process uses strong alkaline to dissolve bauxite, and the resulting red mud waste contains a high content of iron, and its mineral composition mainly includes hematite, perovskite, goethite, calcite, calcite, diaspore and so on [22]. The chemical composition of Bayer red mud mainly depends on the composition of bauxite [23], the production process of alumina and the additives mixed in the production process, which include mainly iron oxide, alumina, silicon dioxide, calcium oxide, in addition to

some trace non-ferrous metals. It was thus found that there were higher contents of $SiO_2$, $Al_2O_3$ and $Fe_2O_3$ than that of other oxides.

**Table 1.** Chemical constituents of Bayer red mud.

| Chemical Composition | $Fe_2O_3$ | $Al_2O_3$ | $SiO_2$ | CaO | MgO | $TiO_2$ | $Na_2O$ | $K_2O$ | Loss |
|---|---|---|---|---|---|---|---|---|---|
| Content/% | 34.3 | 21.4 | 20.1 | 3.2 | 0.3 | 2.0 | 8.1 | 0.2 | 9.7 |

### 2.1.2. Engineering Properties of Red Mud Waste

The Bayer red mud used in this study was sampled on site and the related physical and mechanical properties were tested in laboratory.

- The specific gravity of Bayer red mud ranged between 2.70–3.10;
- The measured value of compression coefficient $a_{1-2}$ ranged between 0.15–0.35 $MPa^{-1}$ which has medium compressibility in accordance with Code for design of building foundation (GB 50007-2011);
- The liquid limit (WL) was more than 50%, and the plasticity index IP was less than 17, which is similar to the engineering characteristics of high liquid limit silty clay in accordance with Test methods of soils for highway engineering (JTG 3430-2020);
- The unconfined compressive strength under the optimal water content state was between 400–600 kPa, and its water stability was poor, leading to disintegration immediately after immersion.

Figure 1 shows the particle size distribution of Bayer red mud. Its particles were very fine, in which the particle content of 1.0–10 μm accounted for more than 90%. The mineral composition of red mud particles with negative charges bears strong hydrophilicity, which makes the thickness of particle water film thick, and the water content can still remain between 35–40% after pressure filtration.

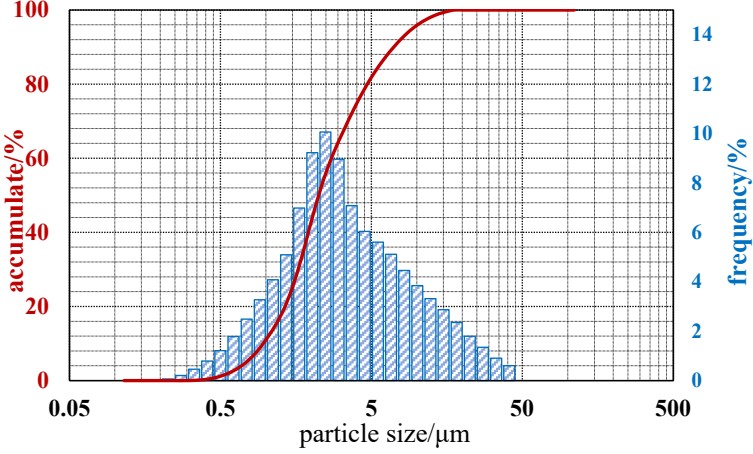

**Figure 1.** Particle distribution curve of Bayer red mud.

It can be concluded that the engineering characteristics of Bayer red mud used exhibited high natural water content, high liquid limit, low strength and poor water stability, which cannot be directly used for road subgrade filling.

### 2.1.3. Leaching of Hazardous Substances of Red Mud Waste

Solid waste substances from production, living and other activities can be divided into solid waste and hazardous waste [24]. According to the test items and methods specified in the General Principles for Identification of Hazardous Wastes (GB 5085.7-2019), the Bayer red mud was tested. It was found that the pH test value ranged from 11.4 to 12.2. The leaching concentrations of detectable hazardous substances are shown in Table 2. The

results showed that leaching concentration of certain harmful elements of Bayer red mud were lower than that of the concentration limit specified in the standard, and should thus not be listed in the national hazardous waste list.

**Table 2.** Leaching concentrations of some harmful elements from Bayer red mud.

| Project | Total Cr | As | Se | Mo | Sb | V | $Cr^{6+}$ | $F^-$ |
|---|---|---|---|---|---|---|---|---|
| Leaching concentration (mg/L) | 1.5 | 0.0097 | 0.01 | 0.16 | 0.03 | 1.10 | 1.45 | 16.1 |
| Limit value of hazardous waste (mg/L) | 15 | 5 | 1 | — | — | — | 5 | 100 |

*2.2. Modification of Bayer Red Mud*

2.2.1. Modifier used in this Study

Due to the poor engineering properties measured above, Bayer red mud requires modification before application. In this study, the solidification and modification treatment adopted a modifier to improve the physical and mechanical properties, to reduce pH value, to inhibit the effect of metal ion leaching, through the action of charge neutralization, adsorption bridging and surface adsorption.

The main components of the modifier comprised cement, phosphogypsum and polymer stabilizer, each component fully mixed in certain mass proportions under normal temperatures. The modifier was then used to treat the Bayer red mud. The cement used was a hydraulic cementing material with 42.5 grade ordinary Portland cement employed. Phosphogypsum is a solid waste produced by a wet-process phosphoric acid process, which was dried and ground to below 0.15 mm before use. The polymer stabilizer is a high molecular stabilizer and water-soluble polymer. This modifier material is a powdery solid at room temperature. After homogenous mixing with Bayer red mud and water, it can be chemically solidified. Cement forms C–S–H gel, $Ca(OH)_2$, Aft and Afm. The pH value of the modified red mud can be adjusted to about 10, which can effectively inhibit the leaching of lead and cadmium. The polymer stabilizer in the modifier can reduce the distance between red mud particles through charge neutralization and form colloidal particle aggregation structure. At the same time, the strong adsorption capacity of the polymer stabilizer effectively connects the dispersed particles to form a whole through "bridging", which not only forms the strength through the crystal network structure skeleton, but also stabilizes the harmful metal ions in the red mud through physical wrapping and chemical adsorption solidification. Compared with traditional lime, cement and other curing materials, addition of modifier can not only improve the strength of red mud, but also improve the water stability of solidified red mud and the diffusion and migration of some metal ions, making it better in actual road performance and environmental sustainability.

2.2.2. Mixture Design and Sample Preparation

The modified red mud mixtures were prepared with five different amounts of modifier at 4%, 6%, 8%, 10% and 12% by mass, respectively. If the content of modifier is lower than 4%, mixing with red mud may not be uniform enough to affect the later modification effect. If the content of modifier is higher than 12%, it will reduce the economic value.

A set of standard Proctor compaction tests in accordance with Test methods of soils for highway engineering (JTG 3430-2020) were conducted to determine optimum moisture content ($\omega$) and maximum dry density ($\rho_d$). The cylinder specimen with a diameter of 39.1 mm and a height of 80 mm was compacted for each condition, and its degree of compaction was 96%. After molding, it was sealed with a sealing bag and placed in a standard curing box at a temperature of $20 \pm 1\,°C$ and a relative humidity of $\geq 90\%$ for a specific number of curing days in the laboratory.

### 2.3. Case Study of a Practical Application of Modified Bayer Red Mud as Subgrade

In order to realize large-scale utilization of modified Bayer red mud in real-world applications, a demonstration road trial was built in a new Ji-Qing Highway Reconstruction and Expansion Project in 2015 in China. This highway was completed in early 1994 and bears a total length of 318 km. In recent years, with the rapid development of regional economy and society and the rapid growth of traffic volume, its actual traffic volume has far exceeded the design traffic volume, thus it is urgent to implement the reconstruction and expansion of engineering projects. In 2015, the Ji-Qing Highway Reconstruction and Expansion Project was officially constructed, from two-way four lanes to two-way eight lanes.

Locally, a number of large alumina enterprises are distributed in Binzhou and Zibo along the reconstruction and expansion project of Jinan Qingdao Highway. For effective usage of locally available resources to minimize cost, a demonstration road trial using the modified red mud was concerned and located on the right side of the stake number (k289 + 090.5~k294 + 385). The subgrade length was about 4900 m, the subgrade width was 11 m, and the filling thickness was 0.2 m. The modified Bayer red mud was used to replace 0.2 m lime-stabilized soil on the upper part of roadbed as shown in Figure 2. The Bayer red mud was evenly mixed with the solidified modifier in the mixing plant, and then transported to the subgrade filling site for paving and compaction. The mass ratio of Bayer red mud and solidified modifier by mass was 92:8. The demonstration road consumed about 17,000 tons of Bayer red mud and saved about 14,800 m$^3$ of natural soil.

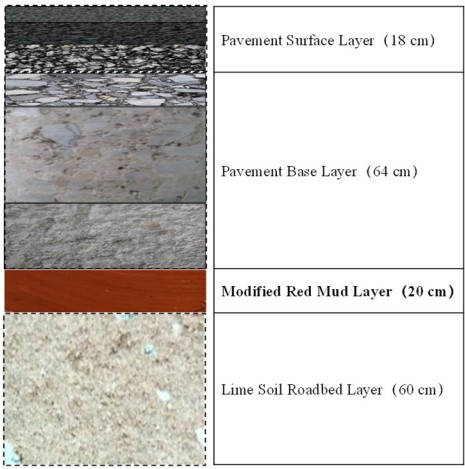

**Figure 2.** Pavement structure diagram of demonstration road.

## 3. Results and Discussion

### 3.1. Properties of Solidified Red Mud in the Laboratory

#### 3.1.1. Effect of Modifier on Optimum Moisture Content and Maximum Dry Density

Figure 3 shows the influence of modifier dosage on the optimal moisture content and maximum dry density of the modified red mud. It was found that the modifier content had a great influence on the maximum dry density of Bayer red mud, while it had less influence on the optimum moisture content. This could be due to a chemical reaction that occurred when the solidified modifier made contact with the Bayer red mud, which would consume a certain amount of water. At the same time, the modifier material allowed red mud particles to more easily form dense agglomerative structures, and its dry density improved from a macroscopic point of view.

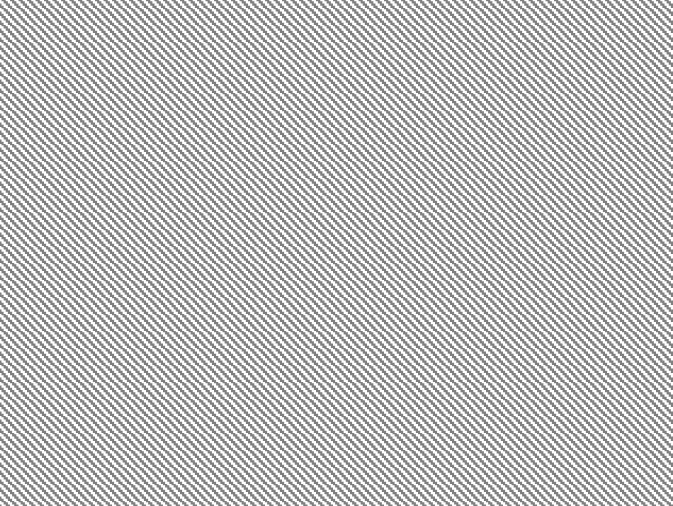

**Figure 3.** Influence of modifier on optimum moisture content and maximum dry density.

3.1.2. Unconfined Compressive Strength of Solidified Red Mud

Figure 4 illustrates the unconfined compressive strength of the modified red mud cured for 3 days, 7 days and 28 days. The unconfined compressive strength test was carried out under the condition of zero lateral pressure, with the application of an axial pressure with an equal axial deformation, until sample failure. It can be observed that with the curing days and the modifier dosage, the strength continued increasing. However, compared to the 28-day strength, the influence of the modifier dosage became insignificant after 8% and the change of strength was slight. This implies that an optimum modifier dosage to treat the red mud waste would be 8%.

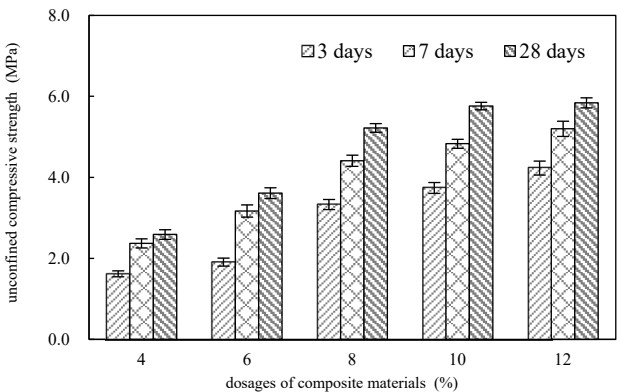

**Figure 4.** Unconfined compressive strength of solidified red mud at different dosages.

After immersion in water for one day, the unconfined compressive strength of the specimens cured for 7 days, and 28 days were tested as well, and the test results are shown in Figure 5. For this immersion test, the specimens were removed from the curing box and soaked in a 20 °C water bath for 24 h. Then, the compressive test was carried out according to the standard method. It was found that after one-day immersion in water, the strength of all specimens had decreased. The level of decrease for the 28 days cured mixture prepared with 8% of modifier was less than 20%.

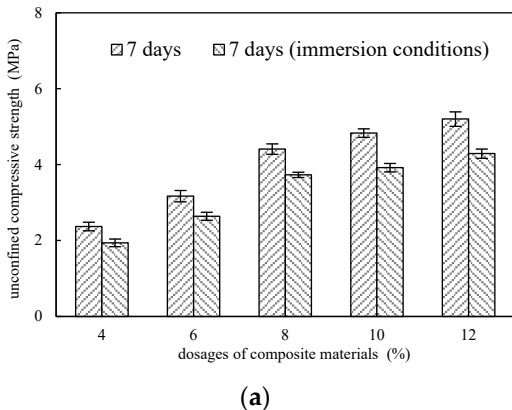
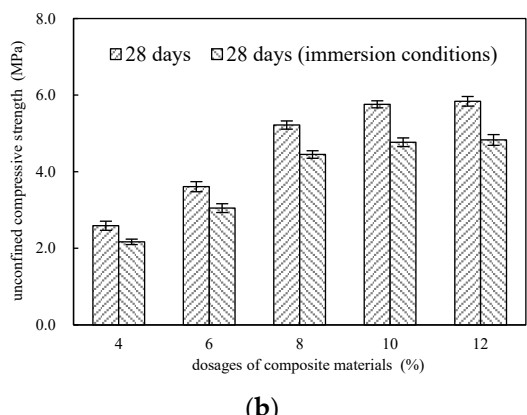

|  (a)  |  (b)  |

**Figure 5.** Unconfined compressive strength of solidified red mud after immersion in water. (**a**) is strength comparison of 7 days, (**b**) is strength comparison of 28 days.

### 3.1.3. Road Performance of Bayer Red Mud before and after Curing Modification

By comparing the results of unconfined compressive strength and strength loss of the modified red mud under standard curing and immersion conditions, the optimal mixing ratio of the modified modifier was determined as 8%. The solidification of Bayer red mud containing 8% modifier was carried out in the laboratory, and the solidification effects on road performance were compared and analyzed. The test results are listed in Table 3. In Table 3, $\omega$ denotes optimum moisture content, $\rho_d$ denotes maximum dry density, $w_L$ denotes liquid limit index, $w_P$ denotes plastic limit index, $I_P$ denotes plasticity index, $\varphi$ denotes internal friction angle, c denotes cohesion, $E_0$ denotes modulus of resilience, CBR denotes California Bearing Ratio.

**Table 3.** Comparison of road performance of Bayer red mud before and after curing modification.

| Category | $\omega$ (%) | $\rho_d$ (g/cm$^3$) | $w_L$ (%) | $w_P$ (%) | $I_P$ | $\varphi$ (°) | c (kPa) | $E_0$ (MPa) | CBR (%) |
|---|---|---|---|---|---|---|---|---|---|
| Original red mud | 27.7 | 1.603 | 51.4 | 38.1 | 13.3 | 24.2 | 31.0 | 27.3 | 4.3 |
| Modified red mud | 28.7 | 1.660 | 37.2 | 22.1 | 15.1 | 36.8 | 143.2 | 705.4 | 124 |

Through the solidification and modification treatment, the liquid limit of the Bayer red mud reduced from 51.4% to 37.2%, the plasticity index increased from 13.3 to 15.1, and the physical properties were improved. At the same time, the shear strength, modulus of resilience and CBR of modified Bayer red mud were significantly improved, and the results showed that the modification treatment could effectively improve the mechanical and road performance of Bayer red mud.

### 3.1.4. Leaching of Hazardous Substances in Solidified Red Mud

Table 4 lists the leaching concentration and reduction ratio of some hazardous substances in the solidified red mud compared with the original one in Table 2. Comparative analysis showed that the pH value of leaching solution decreased from 10–12 to 9–10. The leaching concentrations of total chromium, arsenic, selenium, molybdenum, antimony, vanadium, hexavalent chromium and fluoride decreased significantly, and the reduction ratio of hexavalent chromium, fluoride, selenium, arsenic and vanadium was more than 70%.

**Table 4.** Leaching concentration and reduction ratio of some hazardous substances in solidified red mud.

| Category | Total Cr | As | Se | Mo | Sb | V | Cr$^{6+}$ | F$^{-}$ |
|---|---|---|---|---|---|---|---|---|
| Leaching concentration(mg/L) | 1.21 | 0.0028 | 0.0013 | 0.12 | 0.02 | 0.34 | 0.031 | 2.68 |
| Percentage reduction (%) | 19.3 | 71.1 | 87.0 | 37.5 | 33.3 | 69.1 | 97.9 | 83.4 |

In general, the modified Bayer red mud had a significant effect on inhibiting the leaching of hazardous substances. The modified Bayer red mud can meet the standard requirements of subgrade filling and environmental protection in accordance with specifications for design of highway subgrades (JTG D30-2015) and technical specification of red mud (Bayer) subgrade application for highway engineering (DB 37/T 3559-2019).

*3.2. Performance of Modified Bayer Red Mud as Subgrade in Practical Application*

3.2.1. Construction Method and Quality Control of Subgrade with Modified Bayer Red Mud

In a real project, a centralized plant mixing method was adopted to produce the modified Bayer red mud and to construct the subgrade. The main technological process is given below: preparation before construction → Bayer red mud storage and transportation → preparation of modified materials → construction site leveling and distribution → centralized mixing at mixing station → transportation of modified red mud to site → paving of modified red mud → vibration compaction (strong vibration twice and weak vibration twice) → final compaction → quality inspection → maintenance → next layer construction. This process is used for paving and rolling on the solidified lime soil, which will not adversely affect the lower bearing layer. The construction site of mixing, paving, compacting and curing is shown in Figure 6.

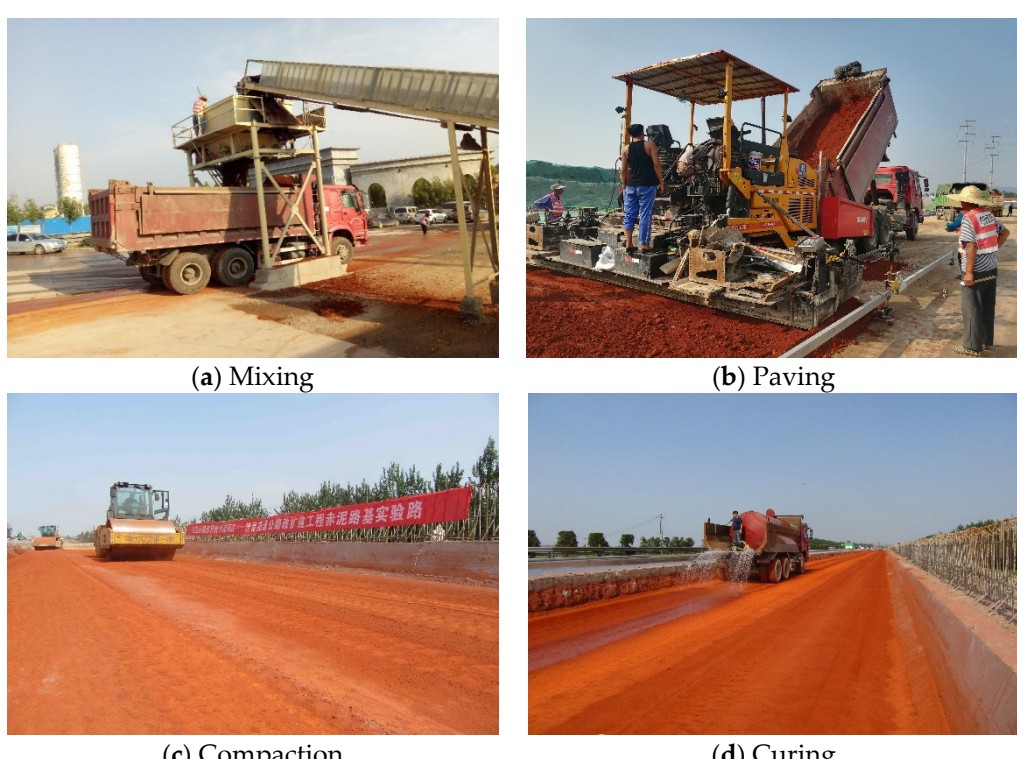

(**a**) Mixing

(**b**) Paving

(**c**) Compaction

(**d**) Curing

**Figure 6.** Plant mixing construction of Bayer red mud in a project.

Regarding the construction, the modified Bayer red mud represents a kind of non-traditional roadbed filling material. Due to its own physical properties and the particularity of the solidification process, some targeted controls on technological aspects were implemented:

The main physical and mechanical properties shall be tested before construction. When the source of red mud waste or the production process changes, these tests must be conducted again:

- The moisture content of the modified Bayer red mud should be tested after mixing in the plant and spreading on site, and the measured moisture content should be controlled at 2–3 percentage points higher than the optimal moisture content.
- The dosage of the solidified modifier should be controlled accurately in the mixing station, and the error should not be more than ±0.5%. After mixing, the surface color of the modified red mud is even and consistent, and there is no ash mass and gray strip. The maximum particle size should not exceed 5 mm.
- In a single-layer construction, the on-site compaction should be completed within 4 h after mixing. In the multi-layer continuous construction, the next filling layer should be compacted within 4 h of the previous filling layer.
- After compaction, the surface should be covered with felt or watered for curing, and the curing period should not be less than 2 days.

3.2.2. Environmental Monitoring of Subgrade Filled with the Modified Red Mud

In the Bayer red mud demonstration road, four sampling wells for groundwater quality monitoring were buried on both sides of the subgrade. The bottom of the sampling wells was below the groundwater level, and the groundwater was sampled regularly [25].

The monitoring well was buried before the Bayer red mud was filled as subgrade, and the groundwater was sampled for testing as a control. After the construction of Bayer red mud, the groundwater in the well was collected regularly at certain interval periods [26]. Figure 7 shows typical indexes of pH, $Cr^{6+}$ and $F^-$ in groundwater samples as a function of time. Through continuous tracking collection and testing, it was found that the pH value and hazardous substance concentrations of groundwater samples in four sampling wells presented different variations. The pH value of groundwater in four places decreased slightly after construction, and finally stabilized in the range of 7.0–8.0. The concentration of hexavalent chromium showed a downward trend from the first blank sampling to 28 days after the completion of the test road, and then measured lower than the limits. The concentration of fluoride showed an upward trend from the first blank sampling to 28 days after the completion of the test road. The concentrations of other metals (total mercury, total chromium, cadmium, lead, beryllium, selenium, silver, copper, antimony, vanadium and cyanide) were lower than that of the detection limits.

According to the comparative analysis of long-term test results above, the pH values and concentrations of hazardous substances in the continuous sampling of groundwater in the test road were lower than that of the limit values of class III groundwater. At the same time, the monitoring results of the test road and the conventional road were consistent overall, and there were no abnormal changes in the concentration of one or more hazardous substances caused by the red mud test road.

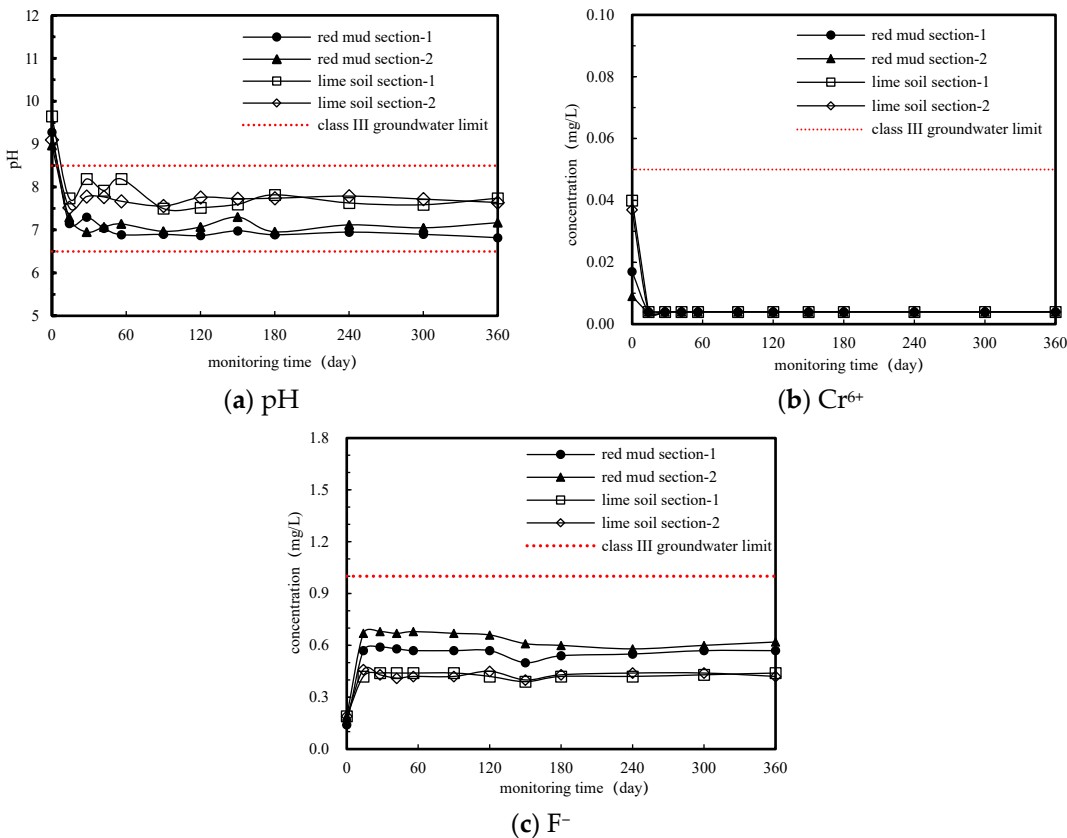

**Figure 7.** Typical indexes pH, Cr$^{6+}$ and F$^-$ in groundwater samples as a function of time.

### 3.2.3. Road Performance of Subgrade Filled with the Modified Red Mud

After seven days of moisturizing and curing in the test road, the compacted Bayer red mud subgrade in the demonstration road was tested on site for road performance. The resilient modulus was measured using a bearing plate method, dynamic modulus of deformation was measured by the portable deflectometer method (PFWD) and the deflection value was measured by the falling weight deflectometer method (FWD) [27–29]. In order to analyze the overall uniformity of subgrade and the correlation of each test index [30], 200 m was selected as the fixed-point test on site, and points were arranged at 20 m intervals in the two lanes of the widening part. The test results are listed in Table 5.

**Table 5.** Performance test results of fixed point test sections.

| Measuring Points | E/MPa | $E_{vd}$/MPa | l/0.01 mm | Measuring Point | E/MPa | $E_{vd}$/MPa | l/0.01 mm |
|---|---|---|---|---|---|---|---|
| 1-1 | 79.4 | 98.2 | 138.9 | 2-1 | 76.9 | 86.0 | 116.2 |
| 1-2 | 108.3 | 166.0 | 89.5 | 2-2 | 143.9 | 217.1 | 49.6 |
| 1-3 | 88.9 | 118.1 | 111.2 | 2-3 | 164.7 | 229.5 | 54.4 |
| 1-4 | 100.7 | 152.3 | 96.9 | 2-4 | 125.2 | 152.0 | 63.1 |
| 1-5 | 110.7 | 156.9 | 81.6 | 2-5 | 174.0 | 260.6 | 50.6 |
| 1-6 | 90.6 | 139.5 | 101.7 | 2-6 | 151.1 | 209.6 | 56.2 |
| 1-7 | 112.3 | 143.5 | 75.4 | 2-7 | 115.4 | 201.1 | 74.0 |
| 1-8 | 134.2 | 187.6 | 74.5 | 2-8 | 169.8 | 199.8 | 62.2 |
| 1-9 | 123.1 | 185.0 | 69.9 | 2-9 | 185.3 | 267.2 | 51.4 |
| 1-10 | 142.3 | 184.4 | 63.7 | 2-10 | 145.7 | 197.9 | 62.5 |

It can be seen from Table 5 that the resilient modulus of most measuring points in the test section was greater than 90 MPa, the dynamic deformation modulus of PFWD was greater than 140 MPa, and the FWD deflection value was less than 100 (0.01 mm). The overall bearing capacity of the subgrade was good and met the road requirements.

Figure 8 further shows correlations between different test indices. These correlations were good enough to provide reference for the quality control and evaluation of the modified Bayer red mud subgrade.

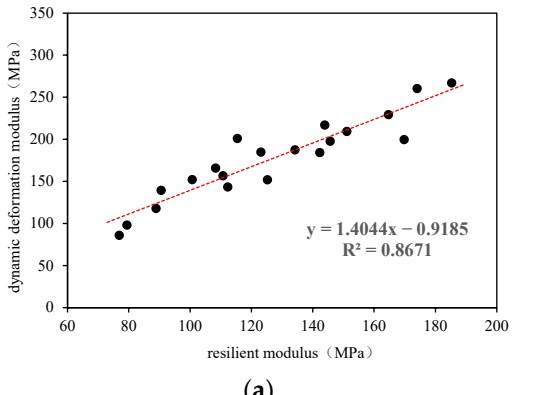

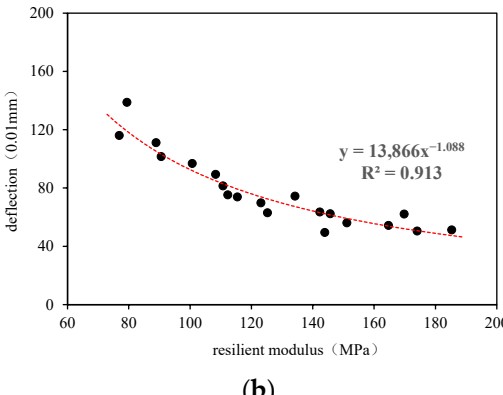

(**a**)

(**b**)

**Figure 8.** Correlation between different performance indexes of modified red mud subgrade. (**a**) is correlation between dynamic deformation modulus and resilience modulus, (**b**) is correlation between deflection and resilience modulus.

Figure 9 shows the cored cylindrical subgrade samples ($\Phi$150 mm $\times$ 150 mm) on site for the unconfined compressive strength test. After testing, the average value of the unconfined compressive strength was 2.4 MPa, which met the specification requirements in this project. In the 95% confidence interval, the unconfined compressive strength of the cored specimen was 2.06–2.77 MPa, and the standard deviation was 0.50. In addition, the white substance on the top surface of the on-site coring specimen comprised $Na_2CO_3$, $CaCO_3$ and other salt substances, which represent the mineral precipitation caused by the humidity change on the top surface of the structural layer exposed to the air.

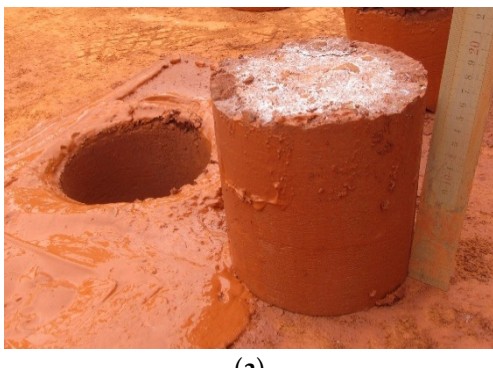

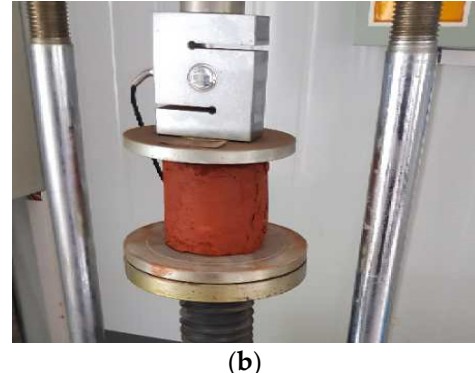

(**a**)

(**b**)

**Figure 9.** Site coring and testing of Bayer red mud Subgrade. (**a**) is field coring sample, (**b**) is unconfined compressive strength test of core sample.

### 3.2.4. Benefit Analysis of Subgrade Filled with the Modified Red Mud

In this practical project, Bayer red mud was used as a local waste resource and its utilization did not require purchasing of materials, with only a certain amount of transportation costs involved. Compared with traditional cement or lime stabilized soil (6%), when the transportation distance was between 15 and 20 km, the modified Bayer red mud needed to fill the subgrade could save CNY 10 per ton compacted earthwork. Through the field test, the representative deflection value of modified Bayer red mud subgrade was less than the representative deflection value of lime stabilized soil, and the modulus was greater than the modulus of lime stabilized soil. This showed that the overall bearing capacity and road performance of Bayer red mud subgrade were better than that of lime stabilized soil subgrade. Therefore, from the analysis of the economic benefits

and utilization of wastes, the use of modified Bayer red mud instead of cement or lime stabilized subgrade is more effective.

Meanwhile, Bayer red mud is now stored in the form of dry-heaping embankment technology, and the enterprise needs to invest approximately CNY 35 per ton in the maintenance and management of the yard every year in China. Among them, the energy cost (gasoline or diesel) of vehicles for loading, unloading, transportation, transportation and dust suppression by watering accounts for about 55%, approximately CNY 19.25 per ton, equivalent to 2.85 L per ton of energy (as calculated by comparison with diesel). Additionally, using one ton of Bayer red mud can directly save approximately 3.49 kg of standard coal per year.

## 4. Conclusions

This study presented the properties of the modified Bayer red mud waste and its large-scale application as subgrade in a real project. The main findings are provided below:

Regarding the leaching results of Bayer red mud, it was identified as a general industrial solid waste by environmental protection standards. However, its engineering characteristics exhibited a high natural water content, high liquid limit, low strength and poor water stability, and thus cannot be directly used for road subgrade filling.

The modifier of Bayer red mud adjusted charge neutralization, adsorption bridging and surface adsorption, and pH value of the microenvironment. Subsequently, a network aggregation structure was formed to reduce the leaching concentration of harmful substances and to improve the mechanical and water stability of the modified Bayer red mud.

Through laboratory testing, the optimal modifier dosage was identified at 8%. The road performance of the modified Bayer red mud revealed a great improvement. The pH value of the leaching solution reduced from 10–12 to 9–10, and the three-day unconfined compressive strength could reach up to 3 MPa. Moreover, the leaching concentration of typical harmful components was reduced by more than 70%.

In the demonstrated engineering application, the resilient modulus of the filled subgrade measured more than 90 MPa, the dynamic deformation modulus more than 140 MPa, and the FWD deflection value less than 100 (0.01 mm). The pH value and hazardous substance concentrations of long-term monitoring groundwater quality samples in the test section were consistent with the monitoring results in the conventional sections and had no adverse effects on groundwater quality.

**Author Contributions:** Data curation, J.W.; Formal analysis, X.Z. and L.Z.; Writing—original draft, S.M.; Writing—review & editing, Z.S. All authors have read and agreed to the published version of the manuscript.

**Funding:** The authors acknowledge the financial support of the National Key R&D Program of China (2018YFB1600103), Shandong Transportation Science and Technology Innovation Project (2016B43) and project ZR2020QE271 supported by Shandong Provincial Natural Science Foundation.

**Institutional Review Board Statement:** Not applicable.

**Informed Consent Statement:** Not applicable.

**Data Availability Statement:** The authors confirm that the data supporting the findings of this study are available within the article.

**Conflicts of Interest:** The authors declare no conflict of interest.

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
