# Peer review of "Utilization of Modified Red Mud Waste from the Bayer Process as Subgrade and Its Performance Assessment in a Large-Sale Application"

_coatings, doi:10.3390/coatings12040471_

Round 1

Reviewer 1 Report

This study presents to adopt a chemical modifier to solidify the Bayer red mud for its real application in highway subgrade. The mechanism and properties of the modified red mud using a modifier, composed of cement, phos-phogypsum and organic polymer, was analyzed and investigated.

Thematically the work is interesting for the researchers and professionals and the proposed manuscript is relevant to the scope of the journal.

I found it appropriate for publication in the Coatings journal, but only after some modifications and clarification from the Authors.

The overall organization and structure of the manuscript are appropriate. The paper is well written and the topic is appropriate for the journal.
The aim of the paper is well described and the discussion was well approached, its results and discussion are correlated to the cited literature data.

The literature review is comprehensive and properly done.

The novelty of the work must be more clearly demonstrated.

The significance of the Work: Given the large number of analyzed data, this is an interesting study with a possible significant impact in this area.

Statistical interpretation of the analytical data must be more properly presented. SD values on figures 3, 4, 5, 7 should be presented?

The verification of the model should be performed. 

Other Specific Comments: The work is properly presented in terms of the language. The work presented here is very interesting and well done, it is presented in a compact manner.

The manuscript should be improved from technical/graphical viewpoint.

Reviewer 2 Report

The authors have identified a relevant subject on utilization of bauxite residue as an alternate geomaterial, thereby conserving the natural resources. The authors have followed most of required tests to be conducted for its utilization as a pavement materials. Authors also have done the quality control tests after of the test section of the road.

However, authors need to add some more points on the manuscript to improved it further.

Authors should check the recent papers on utilization of red mud as an alternate geomaterial with and without stabilization. 

Authors have not discussed about the durability of the developed material in terms of alternate wet dry cycle or other tests. 

Authors also should present some discussion on the white patch as shown in Figure 9, which is is due to efflorescence.

Authors should also mention some terms like RMB, which is a Chinese currency to the international readers. 

Reviewer 3 Report

The subject presented in the paper is sound and significant. Increasing value of red mud (bauxite residue) as industrial waste material from alumina production is still the poblem around the world. It can pose a significant environmental hazard if not stored properly (because of its alkalinity). That aspect of red mud application can be explained more detailed in the introduction. Of course when we modify the red mud we try to change the structure of that material. Very intersting part of the paper is presented case study of application of modified Bayer red mud as 20 cm thickness subgrade. Please add comment if there was any risk of pollution of lime soil roadbed layer during application of modified red mud layer and before compaction of that layer. Was it controlled or checked? In general, reviewed paper is well organized and interesting. The conclusions support the findigs presented in the paper.  

Reviewer 4 Report

The manuscript titled “ Utilization of Modified Red Mud Waste from the Bayer Process as Subgrade and its Performance Assessment in a large-sale application” is an study aiming to  use Bayer red mud as a roadbed filling material. An extensive set of experimental tests have been undertaken, and the proposed approach has been applied to a field investigation. The papers is well written, interesting, with valid discussions. The authors should address the following comments:

Please explain the main application of the red mud mixtures and if these mixtures satisfy the gradation specifications as a filling or subgrade material?

Please elaborate on the curing conditions of the samples (time, temperature, etc). Please  also explain that how the considered curing conditions represent actual field conditions

Please provide more information on the utilized binder for stabilizing the mixtures? What is the advantage of the stabilizer in relation to the available traditional binders (lime, cement, etc) in terms of performance, cost, and environmental sustainability.

Please define the symbols of table 3 in the text.

Please elaborate on the selection of the stabilizer dosages (4-12%). Is there any specific reason for selecting these values?
